# A Promising Approach for the Food Industry: Enhancing Probiotic Viability Through Microencapsulated Synbiotics

**DOI:** 10.3390/microorganisms13020336

**Published:** 2025-02-04

**Authors:** Iuliu Gabriel Malos, Diana Pasarin, Andra-Ionela Ghizdareanu, Bogdan Frunzareanu

**Affiliations:** 1Faculty of Animal Productions Engineering and Management, University of Agronomic Sciences and Veterinary Medicine of Bucharest, 59 Marasti Blvd., District 1, 011464 Bucharest, Romania; gabriel-iuliu.malos@usamv.ro; 2National Research and Development Institute for Chemistry and Petrochemistry—ICECHIM, 202 Splaiul Independentei, 060021 Bucharest, Romania; 3Institute for Control of Biological Products and Veterinary Medicines, 39 Dudului St., 011061 Bucharest, Romania; bogdan.frunzareanu@icbmv.ro

**Keywords:** probiotic, synbiotics, microencapsulation, inulin, pectin oligosaccharides

## Abstract

The role of prebiotics and probiotics in promoting gut health is increasingly recognized in food development and nutrition research. This study explored the enhancement of probiotic viability in the food industry through microencapsulated synbiotics, focusing on *Lactiplantibacillus plantarum* NCIMB 11974 with fructooligosaccharides (FOSs) and inulin as prebiotics. The effect of encapsulation in a chitosan-coated alginate matrix on probiotic survival under simulated gastrointestinal conditions showed a significant effect of 2% FOS concentration on the growth of *Lactiplantibacillus plantarum* NCIMB 11974. The optimization of microencapsulation parameters by the Taguchi method revealed a 2% sodium alginate concentration, a nozzle size of 200 µm, and a concentration of 0.4% chitosan solution as ideal, producing microcapsules with an estimated average diameter of 212 µm. Viability assessments in simulated gastric juice and simulated intestinal juice showed that chitosan-coated alginate microcapsules notably enhanced probiotic survival, achieving log 8 CFU mL^−1^ viability in both environments, a marked improvement over the uncoated variant. The study emphasizes the importance of microencapsulation, particularly by sodium alginate and chitosan, as a viable strategy to improve the survival and delivery of probiotics through the digestive system. By improving the stability and survivability of probiotics, microencapsulation promises to expand the use of synbiotics in various foods, contributing to the development of functional foods with health-promoting properties.

## 1. Introduction

Prebiotics, defined by the Food and Agriculture Organization (FAO) in 2007 as non-viable food components that provide host health benefits by modulating the microbiota, are an emerging field in nutrition research and food development. Commonly known prebiotics include FOS, galactooligosaccharides (GOSs), isomaltooligosaccharides (IMOs), xylooligosaccharides (XOSs), transgalactooligosaccharides (TOSs), and soybean oligosaccharides (SBOSs), as well as inulin and cellulose [1]. These components, mostly carbohydrates with different molecular structures, contribute to body health by promoting the growth and activity of beneficial bacteria in the colon [2]. The main difference between prebiotics and dietary fibers lies in the specificity of their fermentation: prebiotics are fermented exclusively by certain categories of microorganisms, thus providing a selective advantage for targeted probiotic strains [3]. Through their selective fermentation, prebiotics support the proliferation of beneficial gut bacteria, thus contributing to a balanced gut microbiome and overall health.

Probiotic bacteria, also known as lactic acid bacteria, are increasingly recognized for their significant role in promoting health in various food systems. This is mainly due to their biosafety, functional properties, and technological applications [4]. In this context, probiotics, defined as live organisms that provide health benefits to the host when administered in sufficient quantities, are involved in the regulation of gut microbial balance through various mechanisms, including competition with pathogens and stimulation of the immune response [5]. The most commonly used probiotic strains include *Lactobacillus*, *Bifidobacterium*, *Saccharomyces*, *Streptococcus*, and *Enterococcus*. When used together with prebiotics, these combinations offer significant benefits, especially in food systems, by improving survivability during storage and overall functionality. Studies have shown that synbiotics improve the diversity of gut microbiota and increase the production of short-chain fatty acids (SCFAs), which promote systemic health [6,7].

The synergy between prebiotics and probiotics (synbiotics) illustrates the potential of optimized combinations of ingredients to promote gut health. Synbiotics are food ingredients or food supplements that combine probiotics (helpful intestinal bacteria) and prebiotics (indigestible fiber that promotes bacterial growth). More specifically, synbiotics are mixtures of these two components that work together synergistically in the digestive tract [8]. The original definition by Gibson and Roberfroid in 1995 describes synbiotics as mixtures that enhance the survival and colonization of live microbial food supplements in the gastrointestinal tract. These mixtures selectively stimulate the growth and metabolism of health-promoting bacteria [9]. However, the International Scientific Association for Probiotics and Prebiotics (ISAPP) has updated the definition to include a broader range of combinations. According to ISAPP, a synbiotic is a mixture of live microorganisms and one or more substrates that are selectively utilized by the host microorganisms and have a health benefit for the host [10]. This definition emphasizes the synergistic relationship between probiotics, prebiotics, and other substances. Despite the promising health benefits, the widespread use of synbiotics is still hampered by challenges such as practical applications, underutilized analytical tools, conflicting study results, unclear mechanisms, and legal hurdles [11]. The mechanism of action of synbiotics is to modulate the composition and activity of the intestinal microbiota, thereby promoting healthy and diverse microbiota. This is achieved through the simultaneous provision of probiotic microorganisms that can colonize the intestine and prebiotic substrates that promote the proliferation and activity of these and other beneficial native bacteria [12]. In addition to their health benefits, synbiotics are also of great importance for industrial applications in food systems. They improve storage stability and ensure that the probiotics are viable during digestion. These properties demonstrate their potential for scalability and commercial utilization [6].

Synbiotics improve the digestion and absorption of nutrients by facilitating the breakdown of complex carbohydrates and proteins into simpler, more bioavailable forms. The SCFAs produced during prebiotic fermentation contribute to this process by lowering the pH in the colon, creating an inhospitable environment for pathogens, and improving the absorption of important minerals such as calcium and magnesium. Also, synbiotics support immune function, strengthen the integrity of the intestinal barrier, and promote the production of SCFAs, which positively impact systemic health, including anti-inflammatory effects [13,14]. SCFAs, including acetate (C2), propionate (C3), and butyrate (C4), provide significant benefits for gut and systemic health. Butyrate, the primary energy source for colonocytes, strengthens the intestinal barrier, reduces inflammation, and promotes mucosal healing, while propionate supports glucose regulation and lipid metabolism, and acetate promotes systemic metabolic functions. The carbon chain length influences their role, with butyrate having the most direct impact on gut health. The efficacy of synbiotics in SCFA production depends on the prebiotic substrate and probiotic strain, highlighting the need for precise formulation to optimize SCFA profiles and maximize health benefits [15].

The effectiveness of probiotics when used in food is limited by their survival in unfavorable conditions of the human gastrointestinal tract. In this case, despite some disadvantages (such as cost and complexity of the process), microencapsulation technology has gained popularity as it helps to create healthier and tastier foods and is a promising method to protect probiotics from unfavorable gastrointestinal conditions and ensure their targeted release in the colon. Key microencapsulation techniques, materials, and their advantages are summarized in Table 1.

For probiotic applications, it is crucial to use bacterial cultures in their late exponential or stationary phase, as these stages ensure maximum cell density and resistance, which are essential for encapsulation and survival under gastrointestinal conditions. Microcapsules consist of walls of natural or modified polymers that form airtight, permeable, or semi-permeable shells. Over time, these shells break down into non-toxic products through hydrolysis or enzymatic processes [25].

The ideal polymeric material for coating should have the following properties: it shields the active ingredient from light, oxygen, heat, and pH variations; it remains unaffected by the active ingredients; it provides a gradual release of the encapsulated active ingredient; it limits the diffusion of the active ingredient into the external environment; it does not alter the taste; it is convenient for large-scale production; it dissolves in aqueous media, solvents or on melting and is easy to handle.

These encapsulation materials, many of which are generally recognized as safe (GRAS), ensure the viability and functional integrity of probiotics in the gastrointestinal tract while being scalable and industrially feasible [26].

This emphasizes the need for careful selection of biomaterials for the encapsulation of probiotics, taking into account their physicochemical properties, biocompatibility, stability under gastrointestinal conditions, and the procedural nuances of encapsulation techniques. The strategic selection of encapsulation materials such as alginates, carrageenan, xanthan gum, and others plays a critical role in ensuring the viability and functional integrity of probiotic cells in the challenging environment of the gastrointestinal tract [27]. Inulin, a prebiotic, is known to promote the growth of beneficial gut bacteria, especially lactobacilli and bifidobacteria. Both Zhu (2020) and Ni (2020) highlight the potential of inulin to modulate the gut microbiome, with Zhu focusing specifically on its use in dairy calves. However, the choice of inulin for lactobacilli over bifidobacteria could be explored further. In another study, inulin was found to have a stronger growth-promoting effect on *Bifidobacterium animalis subsp*. lactis B420, suggesting that it may be more beneficial for this strain. Inulin with a lower molecular weight was also found to have a better growth-promoting effect on *Lactobacillus*. These results suggest that the choice of inulin for lactobacilli over bifidobacteria may be influenced by the specific strain and the molecular weight of the inulin [28,29,30].

The aim of this study was to investigate the efficacy of prebiotics such as FOS and inulin in promoting the growth of the probiotic *Lactiplantibacillus plantarum* NCIMB 11974. The research also investigated the survivability of synbiotic formulations, both unencapsulated and encapsulated, in gastrointestinal simulants. For encapsulation, an alginate matrix was used, reinforced with a chitosan coating to improve stability and targeted delivery. By exploring these innovative systems, the study emphasizes their potential for industrial applications, particularly in improving the viability of probiotics during storage and digestion, thus contributing to the development of effective synbiotic formulations for gut health.

## 2. Materials and Methods

### 2.1. Materials

This study used probiotic *Lactiplantibacillus plantarum* NCIMB 11974 from the National Collection of Industrial Food and Marine Bacteria, an aerobic bacterium with proven anti-inflammatory properties on the intestinal mucosa [31]. The MRS agar and broth from Carl Roth were used for the growth and development of the probiotic strain. The prebiotics used in all experiments were FOS obtained from BioCare Limited, Birmingham, United Kingdom, and inulin obtained from Dohler, Germany, respectively. For microencapsulation, materials such as sodium alginate and chitosan from Merk, Germany, were chosen due to their gel-forming properties and compatibility with the intestinal environment. The reagents employed for this research were commercially sourced and met analytical grade standards.

### 2.2. Methods

#### 2.2.1. Effect of Prebiotic Concentration of *Lactiplantibacillus plantarum* NCIMB 11974 Growth

The strain *Lactiplantibacillus plantarum* NCIMB 11974 was activated prior to the experiments by inoculation in MRS broth and incubation for 24 h at 37 °C under aerobic conditions. Different concentrations of FOS and inulin were added in tubes with 5 mL MRS broth sterile. The experimental variants included tubes with 1% and 2% concentrations of FOS and 1% and 2% inulin concentrations.

All tubes were inoculated with *Lactiplantibacillus plantarum* NCIMB 11974 and incubated at 37 °C for 24 h in a Nahita incubator, model 629/70. The 24 h incubation time was chosen to ensure that the bacteria reach an advanced exponential or stationary growth phase corresponding to an optimal cell density (OD 600 ≈ 0.8–1.2) suitable for further experimental purposes. This duration was chosen based on preliminary tests, which showed that the bacterial culture consistently reached a concentration of ~10⁹ CFU/mL after 24 h, which was suitable for microencapsulation and further viability testing. To assess the ability of the bacteria to utilize the prebiotics, their growth was monitored throughout the incubation period. The absorbance at 600 nm was measured at regular intervals of 0, 6, 24, and 48 h of incubation [32].

#### 2.2.2. Microencapsulation of Synbiotics

The encapsulation process was carried out in several steps (Figure 1).

Preparation of the bacterial suspensionTo obtain a bacterial cell suspension of 10^9^ CFU/mL, *Lactiplantibacillus plantarum* NCIMB 11974 was incubated in MRS at 37 °C for 24 h under aerobic conditions. The bacterial suspension (~10⁹ CFU/mL) obtained after 24 h in MRS broth was used for encapsulation due to its optimal density and viability to ensure consistent results in further microencapsulation and viability testing. The bacterial cells were separated by centrifugation at 9000 rpm for 5 min at room temperature. The bacterial cells were washed twice with sterile NaCl solution 0.9% (*w*/*v*).

##### Preparation of the Alginate–Prebiotic Mixture

The washed cells are combined with a prebiotic solution of 10% and then 20%, followed by mixing with 20 mL of sodium alginate solution.

##### Microencapsulation Process

Microencapsulation of synbiotics was performed using the spray extrusion method described by Liet al. [33]. The mixture was injected through a nozzle into 0.18 M CaCl_2_ using laboratory microencapsulation equipment developed by our team. The microcapsules were left in the CaCl_2_ solution for 30 min to gel and then washed twice with 0.9% (*w*/*v*) sterile NaCl solution. Also, synbiotics encapsulated in sodium alginate were added to 100 mL of a chitosan solution with continuous stirring on a Heidolph orbital shaker at 100 rpm, 37 °C for 40 min. The resulting microcapsules were washed and stored in 0.9% (*w*/*v*) sterile NaCl solution at 4 °C.

##### Optimization of Microencapsulation Parameters

The optimization study for microencapsulation of synbiotics used the following experimental parameters: sodium alginate concentration, size nozzle, and the chitosan solution concentration with three levels each, as presented in Table 2.

The choice of these specific values was based on preliminary experiments and literature data. Initial screening experiments identified a range of sodium alginate and chitosan concentrations that provided adequate gelling properties, encapsulation efficiency, and probiotic viability. The concentrations for sodium alginate and chitosan) were selected within the range typically reported in similar studies and ensured compatibility with the spray extrusion process [34,35]. The selected nozzle sizes were also chosen to balance droplet formation and the mechanical integrity of the microcapsules, as demonstrated in our initial trials. For optimizing the microencapsulation process of synbiotics was utilized the Taguchi design methodology, as implemented in the MINITAB-18 software (Minitab, LLC, State College, PA, USA). This approach begins by setting a specific target for a process performance indicator, such as viability maintenance. The design variables were identified, each with three different levels. These variables, along with their specified levels, were used to create an orthogonal array that organized the experiments in a structured way. This array outlines the series of experiments to be conducted and allows for the systematic collection of data on the chosen performance indicator. The collected data were then analyzed to determine the impact of each variable on the performance indicator. Through a series of methodically planned experiments, the study provides a comprehensive assessment of how different variables affect the performance indicator. The determination of the values for each variable is based on a thorough understanding of the microencapsulation process and its associated parameters, with the scope of the parameters determining the number of possible values that can be tested. The selection of the appropriate orthogonal array is facilitated by a selection table that helps to identify the correct array for the experimental configuration. The use of the Taguchi algorithm ensures that all variables and their settings are tested consistently.

The design parameters are then determined, and their variation levels are defined to form an orthogonal array. The experiments specified in the array are conducted to collect data on the performance measure. The data are then analyzed to understand how the parameters affect the performance measure. Taguchi’s orthogonal array allows the study of the effects of many parameters on a performance measure in a compressed set of experiments. Determining the variables requires understanding the process and the parameters, with the range of parameters dictating how many values can be tested and how far apart they should be. The appropriate orthogonal array can be selected using the array selection table, which contains the necessary information to find the name of the array and determine where to find it. This array, created using the Taguchi method, ensures that each variable and each set is tested equally.

Nine experimental conditions were created for three factors and three levels according to the Taguchi method, as shown in Table 3.

##### Measurement of Microcapsule Diameter

Although the manual measurement of microcapsule diameters is common practice, it is time-consuming and labor-intensive. Nevertheless, manual measurements are widely accepted in practice and incur minimal costs. Recent advances in image analysis software have paved the way for the automation of these measurements. The manual determination of the average microcapsule diameter (AMD) was performed according to the protocol described by Guess et al. [36] with minor modifications regarding a minimum number of 100 cells per group. To determine the average diameter of the microcapsule, the samples were prepared and mounted on glass slides using a suitable embedding medium.

High-resolution images were captured using an optical microscope equipped with a digital camera. ImageJ (National Institutes of Health and the Laboratory for Optical and Computational Instrumentation, Madison, WA, USA), an image processing software, was used for image analysis. The software was calibrated against a known reference scale, and thresholding techniques were applied to segment the microcapsules from the background. The diameter of each microcapsule was measured using the software’s measurement tools. Statistical analysis was performed to determine the average diameter and distribution of microcapsules within the sample.

#### 2.2.3. Viability Testing of Free and Microencapsulated Synbiotics in Gastrointestinal Simulants

To determine the number of viable probiotics, 1 g of microcapsules was added to 10 mL of phosphate buffer (pH 6.88), shaken at 200 rpm for 10 min to release the probiotics, and counted using the standard plate counting method.

The gastrointestinal simulants were prepared based on the method described by Huang and Adams, with modifications introduced to adapt the protocol to the specific requirements of this study [37]. Simulated gastric juice (SGJ) was prepared using 0.3 g pepsin (specific activity 1:10,000) added to 100 mL of sterile NaCl (0.5% *w*/*v*), and the pH was adjusted to 2–3 using concentrated HCl.

Simulated intestinal juice (SIJ) was prepared using 0.1 g pancreatin (specific activity 4 × USP) and 0.3 g of bile salts added to 100 mL of sterile NaCl (0.5% *w*/*v*), and the pH was adjusted to 8 using 0.4% (*w*/*v*) sterile NaOH solution. The viability of microencapsulated synbiotics in the sodium alginate and sodium alginate–chitosan, respectively, when exposed to gastrointestinal simulants, was determined in vitro following the method described by Mandal et al. [38]. The viability testing of both microencapsulated synbiotics in SGJ follows a similar protocol under the same conditions. For the test of microencapsulated synbiotics, 1 g microcapsules were used. The samples were added to 10 mL of sterile SGJ and SGI, respectively, and homogenized for 1 min by shaking at 150 rpm. The mixtures were incubated at 37 °C with shaking at 150 rpm for 30, 60, and 90 min, respectively.

#### 2.2.4. Statistical Analysis

The data were analyzed using various statistical software packages. In particular, Minitab 20 statistical software (Minitab LLC, State College, PA, USA) was used for overall data analysis. Tukey’s comparison tests were used to identify significant differences (*p* < 0.05) between different experimental results. In addition, analysis of variance (ANOVA) and the Taguchi method was used to analyze the relationships between variables and optimize the microencapsulation process. A statistical analysis was performed to validate the test results. ANOVA was used to assess the influence of the independent variables on microcapsule size and probiotic viability, while Tukey’s test was used to identify significant differences (*p* < 0.05) between the experimental groups.

## 3. Results and Discussion

### 3.1. Impact of Prebiotic Concentration on the Growth of Lactiplantibacillus plantarum NCIMB 11974

The two prebiotics used influenced the growth of *Lactiplantibacillus plantarum* NCIMB 11974 cells differently, as shown in Figure 2.

Compared to the control group, FOS in a concentration of 2% promoted the growth of probiotic cells. These probiotics can utilize FOS, and their growth is stimulated at a concentration of 2%. FOS and inulin at a concentration of 1% and 2% and FOS at a concentration of 1% showed similar growth patterns to the control.

After determining the optimal concentration of the prebiotic with a stimulating effect on the growth of probiotic cells, synbiotics containing probiotic and FOS in a concentration of 2% were produced, microencapsulated, and then tested for probiotic viability determination. Optimizing the prebiotic concentration provides a balance between enhancing bacterial growth and maintaining capsule integrity, both of which are crucial for the efficacy of synbiotics. FOS served as prebiotics that promoted the growth of probiotic bacteria. These complex carbohydrates can pass through our intestines, where they are digested by gut microbes that convert FOS into SCFAs and vitamins. By establishing the ideal concentration for promoting probiotic growth, the study lays the foundation for the development of synbiotic microcapsules for food products with enhanced efficacy. Encapsulation of these synbiotics further ensures their viability, potentially providing novel solutions for promoting gut health and overall wellness.

### 3.2. Optimization of Microencapsulation Parameters by Taguchi Methodology

To determine the optimal parameters for obtaining microcapsules with a diameter of less than 500 μm by the spray extrusion process, tests were carried out using the Taguchi method and the AMD as the main factor. Subsequently, the results obtained for the microcapsule diameters were analyzed using range analysis and statistical methods such as ANOVA, as shown in Table 4 and Table 5. Optimizing the size range of the microcapsules is crucial to ensure adequate protection against gastrointestinal conditions while allowing the effective release of the probiotics at the target site in the intestine. The average diameter of the microcapsules was 230 ± 10.5 µm (mean ± standard deviation), with a size distribution between 150 µm and 250 µm.

The capsule diameter plays an important role in the release kinetics of probiotics, as smaller capsules (<500 µm) provide a larger surface area for nutrient diffusion and targeted release. In this study, the optimized microcapsules (average diameter of 230 ± 10.5 µm) ensure efficient protection during gastrointestinal passage while facilitating timely release at the site of digestion. These results are supported by other studies in which capsules with a diameter between 200 and 300 µm were observed to provide an optimal balance between mechanical stability and release efficiency [39]. Also, microcapsules with smaller diameters have been shown to improve the diffusion of SCFAs and other metabolites, further supporting their role in gut health [40]. Representative images showing the microcapsules and measurements are shown in Figure 3.

Optical microscopy combined with software-based image analysis provides a robust method for quantitatively analyzing the diameter of microcapsules. This method provides valuable insights into the size distribution of microcapsules and facilitates their optimization for specific applications in various fields. The diameter range observed in this study is in line with the results of Li et al. [40], where microcapsules of similar size showed improved protection and controlled release of probiotics under gastrointestinal conditions. In previous studies, microcapsules smaller than 500 μm were shown to allow better diffusion of nutrients and probiotics, ensuring better viability during gastrointestinal passage [41]. The uniform size distribution in this range (150–250 μm) suggests a high degree of process control, which is important for scaling up the encapsulation technique for industrial applications [42].

#### 3.2.1. Range Analysis

In statistics, range analysis refers to the extent of dispersion observed from the minimum to the maximum values within a distribution. It primarily assesses variability and central tendency, providing descriptive statistics to summarize the dataset. The range is computed by subtracting the lowest value from the highest value; a wider range indicates greater variability, whereas a narrower range suggests lower variability (Table 4).

The range analysis shows the extent of the influence of the parameters on AMD (µm) in the following order: B (nozzle size, µm) > A (sodium alginate concentration, %) > C (chitosan solution, *w*/*v*). The analysis showed that nozzle size had the greatest influence on AMD, a finding consistent with previous studies emphasizing the influence of nozzle diameter on droplet formation and subsequent capsule size [43].

This result highlights the importance of optimizing the nozzle size for achieving uniform microcapsules, which directly impacts the viability and release efficiency of probiotics.

#### 3.2.2. Analysis of Variance (ANOVA)

ANOVA should be used to investigate which parameter has a significant influence on AMD (µm). Normally, a high F value indicates which parameter has a significant influence, as shown in Table 5 [44].

The F-test carried out for the three parameters showed that B (1) and C (3) were classified as significant at a *p*-value of less than 0.05, while A (2) was not significant at a significance level of 0.05. Consequently, in both range analysis and ANOVA, sodium alginate concentration (%) and chitosan solution (*w*/*v*) were found to be the most influential factors affecting AMD values. The ANOVA results show that the sodium alginate concentration and the concentration of the chitosan solution significantly influence the AMD (*p* < 0.05), with the sodium alginate concentration showing the highest impact (F = 105.57). The high F value observed for sodium alginate concentration confirms its crucial role in determining microcapsule structure and integrity, consistent with previous results [45]. Alginate is widely known for its ability to form strong gels, and its concentration directly affects the mechanical properties and encapsulation efficiency of microcapsules [46].

These results confirm the important role of these parameters in the optimization of the microencapsulation process. After using the MINITAB-18 software to optimize the parameters, the minimum AMD value was estimated to be 43.78%. After optimizing the synbiotic microencapsulation process, the following optimal parameters were determined: a 2% concentration of sodium alginate solution, a nozzle size of 200 µm, and a concentration of chitosan solution of 0.4 (*w*/*v*). These optimal conditions for the production of microcapsules by the spray extrusion process (sodium alginate concentration of 2%, diameter size of 200 µm, chitosan solution concentration of 0.4 (*w*/*v*)) were confirmed experimentally, and the microcapsules were characterized in terms of AMD values, which were estimated to be 212 µm. Given these results, understanding the size distribution of microcapsules is crucial to further optimize their performance in probiotic delivery.

### 3.3. Viability Testing of Microencapsulated Synbiotics in Gastrointestinal Simulants

After 30 min, 60 min, and 90 min, samples of 1 g of microencapsulated synbiotics in the sodium alginate and 1 g of microencapsulated synbiotics in the sodium alginate and sodium alginate–chitosan, respectively, were extracted, and the number of viable cells was determined by the standard plate counting method. The microcapsules were placed in phosphate buffer (pH 6.88) for 10 min under shaking to solubilize the microspheres, release the probiotics, and determine viable cells by the standard plate counting method.

The results of the viability tests of the microencapsulated synbiotics in SGJ are shown in Figure 4.

The generally accepted minimum physiological number of probiotic bacteria required for efficacy in intestinal juice is between 10^6^ and 10^7^ colony-forming units (CFUs) per milliliter or gram of product. However, the effectiveness of this number can be influenced by the type of carrier. Non-dairy carriers, such as fruit and vegetable juices, have potential but require further research. In addition, the survival of probiotics under simulated gastrointestinal conditions may be influenced by the specific strain and type of juice used as a carrier [47,48]. The viability of probiotics varies depending on the chitosan coating and chitosan concentration. The number of viable cells determined after microencapsulation was 10^8^ CFU/mL. The differences in probiotic viability between the microcapsules coated with 0.1%, 0.3%, and 0.4% chitosan were statistically significant (*p* < 0.05) as determined by Tukey’s post hoc test, with the 0.4% concentration providing the highest survival under both simulated gastric and intestinal conditions.

Microencapsulated synbiotics in sodium alginate had the lowest survival rate at the end of the test interval (log 4 CFU mL^−1^).

The decrease in viability of the cells encapsulated in the sodium alginate matrix was log 2 CFU mL^−1^ higher than for the microencapsulated synbiotics in sodium alginate–chitosan. The application of the chitosan coating and the optimization of its concentration resulted in microspheres with improved survivability (log 8 CFU mL^−1^). The survivability of the microencapsulated synbiotics in sodium alginate–chitosan against the effect of the SGJ was considered as a criterion for the selection of the optimal chitosan concentration, as its very low pH (1–2.5) is very aggressive for probiotics. To be effective, the probiotic must be present in the intestinal juice in a minimum physiological amount of log 7 CFU mL^−1^, which enables adhesion, resistance, and colonization of the intestinal mucosa, as shown in Figure 5. The experimental data lead to the conclusion that the optimized formulation of microcapsules requires the use of a 2% sodium alginate matrix and a 0.4% chitosan coating.

The combination of chitosan and alginate has been shown in various studies to improve cell viability and mechanical properties. The inclusion of alginate in chitosan-based nanoparticles has improved transfection efficiency while reducing toxicity. Researchers have optimized this system for the long-term encapsulation of cells, achieving high viability and improved mechanical properties. Chitosan–alginate films, especially those made with low-molecular-weight chitosan, exhibit good biocompatibility. Studies have shown that the highest viability of encapsulated *Bifidobacterium pseudocatenulatum* G4 was achieved with a combination of high alginate and low chitosan concentrations. Overall, these results suggest that the chitosan–alginate combination outperforms alginate alone, which may be due to higher transfection efficiency, lower toxicity, and better mechanical properties [49,50,51,52]. The density of the probiotic encapsulated in sodium alginate decreased drastically during the 60 min incubation in the intestinal juice and reached a viability of log 2 CFU mL^−1^, which was far below the therapeutic value. For the microcapsules in the alginate matrix, a loss of viability to a value of log 5 CFU mL^−1^ was observed at the end of incubation. Alginate–chitosan microspheres protected the cells best, with a survival rate above the physiological effective value. The best formulation, which provided viability of log 8 CFU mL^−1^ in simulated intestinal juice, consists of probiotic microencapsulation in a sodium alginate matrix coated with 0.4% chitosan. The improved survivability observed in microcapsules with a 0.4% chitosan coating is consistent with studies showing that chitosan provides an additional protective barrier against low pH and digestive enzymes [53]. It has previously been reported that using sodium alginate as the primary matrix and chitosan as a coating synergistically improves the mechanical stability and viability of probiotics, especially under harsh conditions in the stomach [54]. This study presents a novel approach to optimize synbiotic microencapsulation using a sodium alginate–chitosan matrix, which significantly improved the viability of *Lactiplantibacillus plantarum* NCIMB 11974 (log 8 CFU mL^−1^) under simulated gastrointestinal conditions. By applying the Taguchi method, optimal parameters (2% alginate, 0.4% chitosan, 200 µm nozzle) were determined to produce uniform microcapsules (~212 µm) and provide a scalable solution for functional food applications. Furthermore, the results align with previous studies showing that the viability of microencapsulated synbiotics in sodium alginate decreases over time, while the addition of a chitosan coating significantly improves stability during prolonged incubation. 

## 4. Conclusions

Key findings of this study include the reveal that FOS at a concentration of 2% significantly promotes the growth of *Lactiplantibacillus plantarum* NCIMB 11974. The optimal microencapsulation formulation, identified as a 2% sodium alginate matrix with a 0.4% chitosan coating, achieved high probiotic viability of log 8 CFU mL^−1^ under simulated gastrointestinal conditions. The selection of encapsulation materials and methods is crucial for maintaining the viability of probiotics without compromising the sensory properties of food. Coating composition plays a crucial role in improving the functionality and performance of microcapsules, which further supports the application of microencapsulated synbiotics in the development of functional foods.

From a market perspective, these findings provide a basis for developing scalable and cost-effective production methods for microencapsulated synbiotics, potentially reducing production costs and enabling their inclusion in a wide range of consumer products. Economically, using widely available and affordable materials such as sodium alginate and chitosan improves the feasibility of introducing this technology into large-scale production, benefiting both manufacturers and end consumers. From a nutritional perspective, using synbiotics with optimized prebiotic and probiotic combinations can address specific nutritional needs, such as improving gut health, enhancing nutrient absorption, and supporting immune function in diverse populations. From an environmental perspective, the biodegradability of alginate and chitosan as an encapsulation material meets the growing demand for sustainable and environmentally friendly food packaging and delivery systems. Future research should address the challenges identified in this study by evaluating the stability and functionality of microencapsulated synbiotics in various food matrices, including carbonated beverages, baked goods, and desserts. It is crucial to investigate how these synbiotics interact with other food components to ensure not only their optimal performance but also their sensory acceptability to the consumer. Furthermore, research into different prebiotic and probiotic combinations is essential to uncover synergistic interactions that improve the overall efficacy of microencapsulated synbiotics.

While this study highlights the potential of alginate-based microencapsulation for the protection of probiotics under gastrointestinal conditions, the stability of these formulations in complex and dynamic food systems remains an important area for further investigation. These findings will contribute to the development of robust, scalable formulations that can maintain the functionality of probiotics in various applications and ultimately extend the reach of microencapsulated synbiotics in the industry and to consumers. These results suggest a promising future for the integration of microencapsulated synbiotics into functional foods and nutraceuticals to meet consumer demand for health-oriented and convenient products. The scalability of this technology could also open opportunities for its application in emerging markets and provide access to health-promoting products worldwide.

## Figures and Tables

**Figure 1 microorganisms-13-00336-f001:**
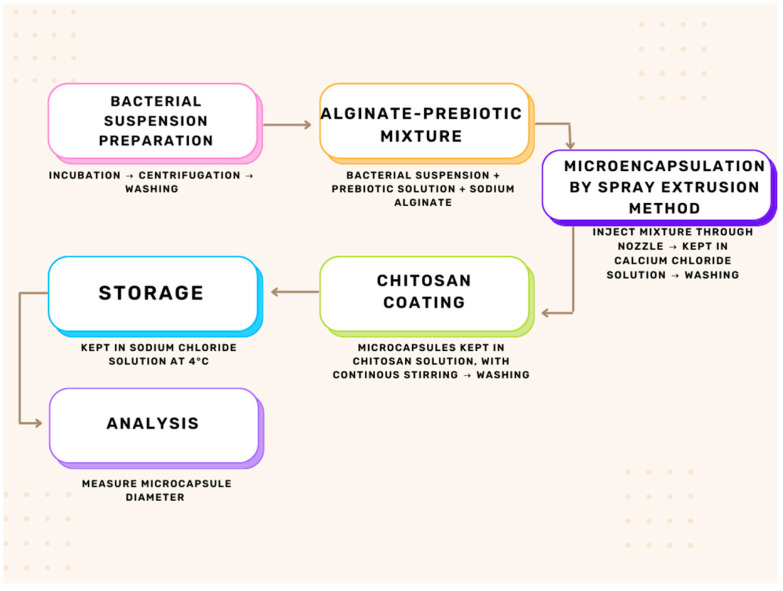
Microencapsulation process steps.

**Figure 2 microorganisms-13-00336-f002:**
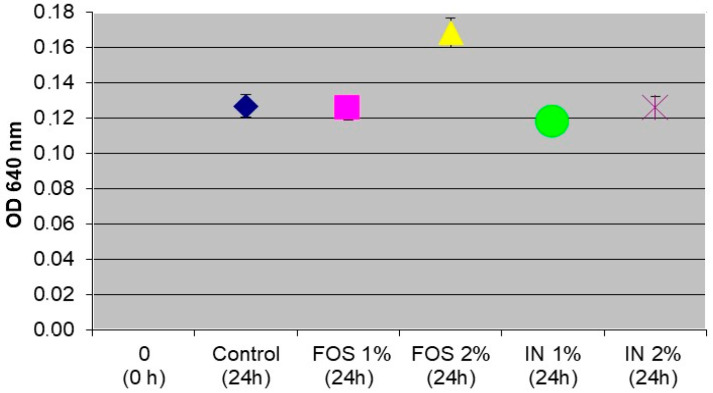
Growth of *Lactiplantibacillus plantarum* NCIMB 11974 in the presence of prebiotics FOS and inulin, respectively.

**Figure 3 microorganisms-13-00336-f003:**
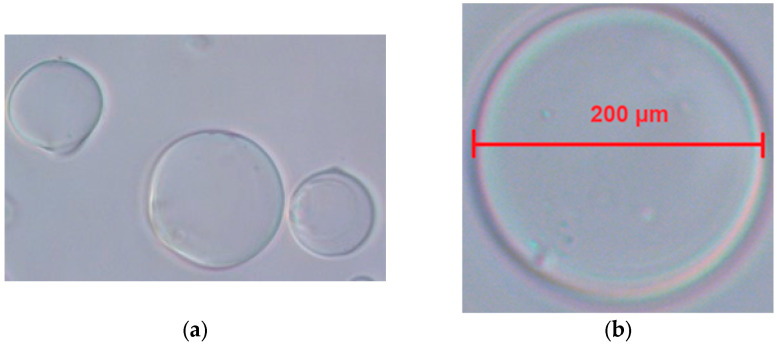
Average diameter of the microcapsule measurements: (**a**) microcapsules; (**b**) reference scale and measurements of a microcapsule.

**Figure 4 microorganisms-13-00336-f004:**
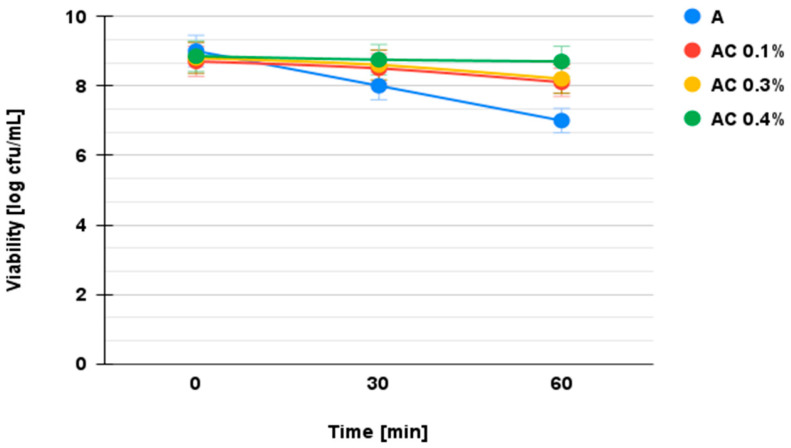
Viability of probiotic *L. plantarum* NCIMB 11974 in SGJ (A—microencapsulated synbiotics in sodium alginate 2%; AC 0.1%—microencapsulated synbiotics in sodium alginate–chitosan 0.1%; AC 0.3%—microencapsulated synbiotics in sodium alginate–chitosan 0.3%; AC 0.4%—microencapsulated synbiotics in sodium alginate–chitosan 0.4%).

**Figure 5 microorganisms-13-00336-f005:**
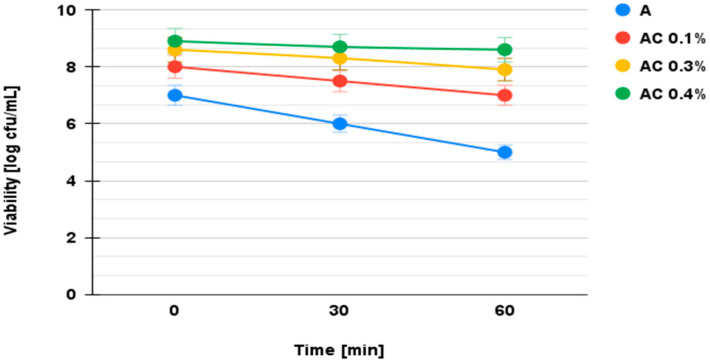
Viability of probiotic *L. plantarum* NCIMB 11974 in SGI (A—microencapsulated synbiotics in sodium alginate 2%; AC 0.1%—microencapsulated synbiotics in sodium alginate–chitosan 0.1%; AC 0.3%—microencapsulated synbiotics in sodium alginate–chitosan 0.3%; AC 0.4%—microencapsulated synbiotics in sodium alginate–chitosan 0.4%).

**Table 1 microorganisms-13-00336-t001:** Summary of encapsulation techniques, materials, and their advantages in probiotic applications.

Type of Capsules	Materials Used	Size Range	Advantages	References
Microcapsules	Alginate, chitosan, carrageenan, xanthan gum	1–5000 µm	Protects probiotics during digestion, controlled release, and improved stability.	[16,17]
Nanocapsules	Gelatin, starch, phospholipids	<1 µm	Suitable for targeted delivery, enhanced bioavailability, and long shelf life.	[18,19]
Microspheres	Polymers, natural biopolymers	1–250 µm	Homogeneous or heterogeneous encapsulation with enhanced probiotic protection.	[20,21,22]
Coated capsules	Lipids, waxes, pectin	Variable (based on core size)	Provides additional layers of protection and ensures gradual release.	[23,24]

**Table 2 microorganisms-13-00336-t002:** Independent variables and their levels within the Taguchi design methodology.

Level	Sodium Alginate Concentration (%)	Size Nozzle (µm)	Chitosan Solution (*w*/*v*)
1	1	200	0.1
2	2	250	0.3
3	2.5	300	0.4

**Table 3 microorganisms-13-00336-t003:** Experimental design for microencapsulation of synbiotics.

Run	A	B	C
R1	1.0	200	0.1
R2	1.0	250	0.3
R3	1.0	300	0.4
R4	2.0	200	0.3
R5	2.0	250	0.4
R6	2.0	300	0.1
R7	2.5	200	0.4
R8	2.5	250	0.1
R9	2.5	300	0.3

**Table 4 microorganisms-13-00336-t004:** The range analysis for the AMD µm was obtained for the L9 matrix.

Run	Parameters			AMD (µm)
	A	B	C	
R1	1.0	200	0.1	275 ± 0.15
R2	1.0	250	0.3	290 ± 0.22
R3	1.0	300	0.4	245 ± 0.31
R4	2.0	200	0.3	260 ± 0.17
R5	2.0	250	0.4	215 ± 0.25
R6	2.0	300	0.1	265 ± 0.36
R7	2.5	200	0.4	220 ± 0.18
R8	2.5	250	0.1	255 ± 0.55
R9	2.5	300	0.3	280 ± 0.34
K (1)—AMD	270.0	251.7	265.0	
K (2)—AMD	246.7	253.3	276.7	
K (3)—AMD	251.7	263.3	226.7	
R—AMD	23.3	11.7	50.0	

The data presented are the average values of measurements taken in three separate experiments. The variables analyzed include (A) sodium alginate concentration (%), (B) nozzle size (µm), and (C) chitosan solution (*w*/*v*). All variables were analyzed according to the obtained AMD. Ki is the sum of the AMD values at level ’i’ within the same column divided by 3. The symbol “R” denotes the difference between the highest and lowest values of K (1), K (2) and K (3).

**Table 5 microorganisms-13-00336-t005:** The ANOVA was performed with the AMD values obtained from the orthogonal matrix L9 (3^3^).

Parameter	DF	Adj SS	Adj MS	F	*p*
Sodium alginate concentration (%)	2	905.56	452.78	23.29	0.041
Nozzle size (µm)	2	238.89	119.44	6.14	0.140
Sodium alginate concentration (%)	2	4105.56	2052.78	105.57	0.009
Residual error	2	38.89	19.44		
Total	8	5288.89			

DF—degree of freedom; Adj SS—adjusted sums of squares; Adj MS—adjusted mean squares; F—value; *p*–value.

## Data Availability

No new data were created or analyzed in this study.

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
