# Peer review of "A Promising Approach for the Food Industry: Enhancing Probiotic Viability Through Microencapsulated Synbiotics"

_microorganisms, 2025, doi:10.3390/microorganisms13020336_

Round 1
Reviewer 1 Report
Comments and Suggestions for Authors
In this work, the authors discuss a promising approach for the food industry, specifically enhancing probiotic viability through microencapsulated synbiotics. The topic is interesting and the manuscript has a potential from a scientific point of view. However, there are some points that require the authors’ attention.
1. Introduction is in general well-written and covers the rationale of this study. I would prefer to narrow down certain parts. For instance, the excess technicalities (about process optimization) in lines 113-145 could be kept to minimal or maybe it would be better to present these in a table. In this way, the authors can create some space in order to focus on available microencapsulation techniques currently applied in the industry. Also, in the same context, lines 76-84 can be linked, so the authors would expand to the advantages of symbiotics (prebiotics and probiotics) especially in terms of utilization in food systems and survivability during storage (e.g. https://doi.org/10.3390/app13010630). In my opinion, these are important aspects for the possibility of scaling up, so they should be addressed effectively.
2. The aim paragraph is small and precise. Maybe the final aim of this work can be further highlighted to better illustrate the rationale of the paper (I leave this one on the authors’ judgement).
Materials and methods is detailed.
3. Line 165: Add a few details about the conditions used for strain activation.
4. An appropriate reference should be added in subsection 2.2.1.
5. Maybe 2.2.2. could be further divided to smaller subsections (if possible). This way it would be easier for the reader to locate specific details. In fact, a flow chart summarizing the whole procedure would be useful, too.
6. Table 1: It is not clear (here or in the discussion) why the authors chose the concentrations described here. Were there any preliminary experiments included? Please elaborate.
Results and discussion:
7. Despite having some interesting results, the whole “Results & Discussion” section (specifically lines 284-338) is mostly oriented to describe results and no real discussion is provided. The discussion should be expanded so that the authors’ findings (and their importance) are linked and compared to the existing literature.
8. Especially, the viability improvements (0.4% chitosan) are highlighted in-text, but no effective comparison to other studies was provided. The role of capsule diameter in probiotic release should also be better covered.
9. I like the outline of the conclusions section. I would prefer the authors to add a few more lines and emphasize about the market/economical, environmental (if any), nutritional, etc insights gained from this work, in order to recommend potential scaling of this work in future.
Author Response
Dear Reviewer,
Thank you for your valuable feedback. We have made all the necessary changes as recommended. Please review both the updated manuscript and the attached response letter.
Please see the attachment.
Best regards,

Reviewer 2 Report
Comments and Suggestions for Authors
The manuscript under evaluation is interesting as the role of probiotics and prebiotics has been well studied the last years given their significant impact as nutritious vehicles against chronic diseases or pathophysiological disorders. The paper is well written, however some modifications in the English language are required. In addition, the authors must provide a list with the most imprortant findings of the study (suggested in the Conclusion section) and discuss in more details how the synbiotics benefit the disgestive tract. Furthermore, they could discuss what are the benefits of short chain fatty acids and if the carbon chain affects the synbiotics role. Other comments that highlight technical errors are given in the attached pdf.
Based on the overall impact of this study. I suggest a minor revision prior to further consideration for publication.

Author Response

(The authors gave the same response as above.)

Reviewer 3 Report
Comments and Suggestions for Authors
In my opinion, the manuscript entitled A promising approach for the food industry: enhancing probiotic viability through microencapsulated synbiotics by Malos et al., aimed to study the efficacy of prebiotics such as FOS and inulin to enhance the growth of the probiotic Lactiplantibacillus plantarum NCIMB 11974. Introduction provides enough information regarding the current state of the art, materials and methods are enough described and results are commented and compared with other studies.
I have some comments and suggestions, as follows:
1. When an abbreviation is used for the first time its meaning must be mentioned under brackets. For instance, FOS from the abstract or FAO in the introduction part.
2. Lines 52-53: the name of the microorganism must be written in italic.
3. Authors must better emphasize the novelty of the study. Lines 430-432 showed that results are consistent with other works. Therefore, which is the novelty of the study?
4. I would like to see an author’s critical point of view of the present work.
5. The conclusion part must highlight only the obtained results and further studies. There is no need to insert in the conclusion part information about microencapsulation.

Author Response

(The authors gave the same response as above.)

Round 2
Reviewer 1 Report
Comments and Suggestions for Authors
The authors have addressed my comments.